# The Effects of Repeated Thermal Stress on the Physiological Parameters of Young Physically Active Men Who Regularly Use the Sauna: A Multifactorial Assessment

**DOI:** 10.3390/ijerph182111503

**Published:** 2021-11-01

**Authors:** Robert Podstawski, Krzysztof Borysławski, Andrzej Pomianowski, Wioletta Krystkiewicz, Tomasz Boraczyński, Dariusz Mosler, Jacek Wąsik, Jarosław Jaszczur-Nowicki

**Affiliations:** 1Department of Tourism, Faculty of Geoengineering, Recreation and Ecology, University of Warmia and Mazury in Olsztyn, ul. Oczapowskiego 5, 10-719 Olsztyn, Poland; j.jaszczur-nowicki@uwm.edu.pl; 2Institute of Health, Angelus Silesius State University, ul. Zamkowa 4, 58-300 Wałbrzych, Poland; krzysztof.boryslawski@upwr.edu.pl; 3Department of Internal Diseases with Clinic, University of Warmia and Mazury in Olsztyn, ul. Oczapowskiego 2, 10-719 Olsztyn, Poland; apomian@uwm.edu.pl (A.P.); wioletta.krystkiewicz@uwm.edu.pl (W.K.); 4Department of Health Sciences, Olsztyn University, ul. Bydgoska 33, 10-041 Olsztyn, Poland; boraczynskitomasz@gmail.com; 5Department of Kinesiology and Health Prevention, Jan Dlugosz University in Częstochowa, 42-200 Częstochowa, Poland; d.mosler@ujd.edu.pl (D.M.); j.wasik@ujd.edu.pl (J.W.)

**Keywords:** Finnish sauna, morphological parameters, biochemical characteristics, fluids, electrolyte levels

## Abstract

The aim of this study was to determine the effects of thermal stress (TS) on changes in blood biochemical parameters and fluid electrolyte levels in young adult men with moderate and high levels of physical activity. Thirty men (22.67 ± 2.02 years) were exposed to four 12-min sauna sessions (temperature: 90–91 °C; relative humidity: 14–16%) with four 6-min cool-down breaks. The evaluated variables were anthropometric, physiological, and hematological characteristics. The mean values of HR_avg_ (102.5 bpm) were within the easy effort range, whereas HR_peak_ (143.3 bpm) values were within the very difficult effort range. A significant increase was noted in pO2 (*p* < 0.001), total cholesterol (*p* < 0.008), HDL (*p* < 0.006) and LDL cholesterol (*p* < 0.007). Significant decreases were observed in the SBP (by 9.7 mmHg), DBP (by 6.9 mmHg) (*p* < 0.001), pH (*p* < 0.001), aHCO_3_- (*p* < 0.005), sHCO3- (*p* < 0.003), BE (ecf) (*p* < 0.022), BE (B), ctCO_2_ (for both *p* < 0.005), glucose (*p* < 0.001), and LA (*p* < 0.036). High 72-min TS did not induce significant changes in the physiological parameters of young and physically active men who regularly use the sauna, excluding significant loss of body mass. We can assume that relatively long sauna sessions do not disturb homeostasis and are safe for the health of properly prepared males.

## 1. Introduction

Sauna bathing is a form of passive heat therapy that has been used for millennia for relaxation, hygiene, health, social and spiritual purposes in many parts of the world [1,2]. Sauna use is particularly widespread in Scandinavia, but it has also gained considerable popularity in other countries, where weekly sauna sessions are regarded as an effective treatment for health improvement [3]. Modern-day saunas include the traditional Finnish-style sauna, Turkish-style hamman sauna, Russian banya and other cultural variations that differ in the design of the heating source and humidity [4].

Sauna use has been extensively studied, and there is scientific evidence to indicate that thermal stress induces similar hemodynamic and endocrinal changes to those evoked by physical exercise [1,5,6]. There are many indications for sauna therapy, and studies investigating the benefits of and contraindications to sauna bathing have been conducted in Finland since the late 1970s [7,8]. In the last three decades, innovative experiments examining new therapeutic uses of sauna bathing have been performed in many other countries on infants and small children [9,10], adolescents [7], young adults [11,12,13] and older adults [14].

The exposure to high temperature and low humidity in a sauna exerts physiological effects on bodily systems and organs and activates thermoregulatory mechanisms that induce reactive changes in the body [5]. The exposure to high (heating phase) and low (cooling phase) temperatures stimulates different processes in the human body. During the heating phase, the cardiovascular system is exposed to thermal stress which causes vasodilation, increases blood perfusion and tachycardia, and stimulates perspiration [15]. The heart rate (HR) of sauna bathers has been shown to double in response to heat; cardiac output can increase by 70% relative to rest values, whereas peripheral vascular resistance decreases by approximately 40% [16]. A decrease in diastolic blood pressure and mean arterial pressure is also observed, whereas systolic blood pressure remains relatively invariant [5,17]. Prolonged exposure to elevated ambient temperature results in the accumulation of heat in the body regardless of the effectiveness of the thermoregulatory mechanisms, which is reflected by the gradual increase in internal body temperature [18]. The high temperature in a sauna leads to changes in the cardiac conduction system. In consequence, skin temperature ranges from 40 °C during the heating phase to 33 °C during the cooling phase after immersion in cold water [19]. Other researchers found that rectal temperature increased by 1.78 °C and tympanal temperature increased by 1.33 °C in men after 30 min of sauna bathing [20]. In a study by Pilch et al. [18], a 30-min session in a dry sauna raised rectal temperature from 37.1 to 38.4 °C and tympanal temperature from 36.8 to 39.3 °C (duration of sauna: 30–40 min, temperature: 80 °C, relative humidity: 2–5%). The strain on the heart and the cardiovascular system is significantly influenced by temperature, air humidity, and duration of stay in the sauna [5], as well as individual traits such physical activity levels, exercise capacity, frequency of sauna use, and general physical and mental health [4].

The exposure to thermal stress decreases body mass [18], mainly due to the loss of total body water [21]. The loss of bodily fluids has been estimated at 0.23–2.3 L, depending on the intensity of the physiological processes induced by thermal stress and duration of heat exposure [22]. Thermal stress induces hormonal changes, and the hormonal system exerts a strong influence on the thermoregulatory system to maintain the body’s thermal balance [3,23,24].

A review of the literature revealed that despite a relatively large number of papers describing the effects of thermal stress on the human body, the vast majority of studies have investigated sauna sessions lasting 30 min or less when the temperature exceeded 80 °C [6,18,25,26,27,28]. There is a general scarcity of published data concerning the effects of repeated sauna use on the physiological profiles of healthy and physically active men who regularly use the sauna. It can be assumed that regular sauna users [29] respond differently to thermal stress than men who use the sauna for the first time or sporadically [30]. Even fewer studies have investigated prolonged or repeated thermal stress lasting more than 40 min at a temperature of 90 °C or higher [7]. These thermal conditions can be regarded as extreme [2], and tolerance to extreme thermal stress has not been fully elucidated in subjects characterized by different levels or physical activity or frequency of sauna use. Studies conducted on obese and sedentary men who were sporadic sauna users demonstrated that repeated use of Finnish sauna induced significant changes in physiological parameters and that these changes were intensified during successive treatments. Deleterious cardiovascular adaptations were most prevalent in men characterized by the highest degree of obesity and largest body size [7]. However, in studies exploring the influence of repeated thermal stress on the human body, the analyzed indicators were limited to the loss of body mass (mainly body fluids), HR, blood pressure (BP) and selected hormones [7,26,29], whereas homeostasis is controlled by numerous organs and systems [31] whose effectiveness can be monitored based on biochemical parameters and fluid electrolyte levels [32,33,34].

Therefore, the aim of the present study was to evaluate the short-term effects of repeated sauna sessions on the physiological parameters of young, healthy, and physically active men who regularly use the sauna.

## 2. Materials and Methods

### 2.1. Participant Selection

The study was conducted in 2020 on 30 male volunteers aged 19–26 years (mean age: 22.67 ± 2.02). The potential participants were informed about the purpose of the study. A total of 47 men volunteered for the study, were notified by e-mail and text message whether they had met the inclusion criteria and were provided with the date of the final recruitment. Before the study, the recruited volunteers attended 10 sauna sessions (twice a week) to prepare for the physiological strain resulting from exposure to high temperature. Each session involved three 10-min stays in the sauna (temperature: 80 °C, relative humidity: 14–16%) with 6-min breaks in between.

The participants were asked to complete a health questionnaire before the study. Based on the results, 30 men meeting the inclusion criteria were recruited for the study. The participants confirmed that they did not take any medications or nutritional supplements, were in good health, and had no history of blood diseases or diseases affecting biochemical and biomechanical factors. None of the evaluated participants had respiratory or circulatory ailments. Their physical activity (PA) levels were evaluated using the standardized and validated International Physical Activity Questionnaire (IPAQ) [35]. The IPAQ was used only to select a homogenous sample of male students, and the results were presented only in terms of Metabolic Equivalent of Task (MET) units indicative of the participants’ PA levels. The MET is the ratio of the work metabolic rate to the resting metabolic rate, and 1 MET denotes the amount of oxygen consumed in 1 min, which is estimated at 3.5 mL/kg/min. Based on the self-reported PA levels in the short IPAQ form “last 7 d recall” [36], 21 of the tested subjects were characterized by moderate PA levels (>600 METs per week) and nine respondents by high PA levels (>1500 METs per week). Only the subjects with moderate and high levels of PA (energy expenditure over 600 METs per week) were chosen for the study.

### 2.2. Ethical Approval

The study was conducted upon the prior consent of the Ethics Committee of the University of Warmia and Mazury in Olsztyn (No. 10/2020), Poland. The study was performed on volunteers who signed an informed consent statement.

### 2.3. Research Protocol

The participants received comprehensive information about sauna rules during the meeting preceding the study. They were asked to drink at least 1 L of water on the day of the test and 0.5 L of water 2 h before the session. The participants did not consume any foods or other fluids until after the final body measurements.

All participants visited a dry sauna in the same location and over the same period (between 8:00–10:00) to minimize the effect of diurnal variation on the results. Every participant attended four sauna sessions (temperature: 90 °C; relative humidity: 14–16%) of 12 min each and remained in a sitting position during each session. After every 12-min session, students recovered in a neutral room (temperature of 18 °C and relative humidity 40–50%) in a sitting position. Each recovery session lasted 6 min, during which the participants remained in a cold paddling pool (water temperature: 10–11° C) for 1 min. Air temperature and humidity inside the sauna chamber and the neutral room, and the temperature of water in the paddling pool were measured with the Voltcraft BL-20 TRH + FM-200 hygrometer (Germany) and confirmed with the Stalgast 620711 laser thermometer (Poland).

Body height was measured to the nearest 1 mm with a calibrated Soehlne Electronic Height Rod 5003 (Soehlne Professional, Germany), and body mass was measured to the nearest 0.1 kg; both values were used to determine the BMI. Blood pressure (BP) and heart rate HR were measured with an automatic digital blood pressure monitor (Omron M6 Comfort. Japan) immediately before the first session and immediately after the fourth cool-down break (72nd minute of the experiment: 72-ME) in the neutral compartment. Due to high temperature in the sauna, physiological parameters, including heart rate (HR _min, avg, peak_), recovery time, energy expenditure, estimated values of oxygen uptake (VO_2 avg, max_), excess post exercise oxygen consumption (EPOC _avg. peak_), respiratory rate (RR _avg, peak_) and physical effort (easy, moderate, difficult, very difficult, and maximal), were measured indirectly using telemetry heart rate monitors (Ambit3 Peak, Suunto Sapphire, Vantaa, Finland) which are widely used in studies of the type [30,37]. Throughout the experiment, the values of physiological parameters were estimated between the first entry to the sauna and the end of the fourth 12-min sauna session (66th minute of the experiment: 66-ME). Suunto heart rate monitors were placed on the wrist, and HR monitor sensors were attached to the chest. Every heart rate monitor was calibrated to male sex, year of birth, body mass and PA level before sauna exposure.

Body temperature was measured on the forehead with a contactless digital thermometer (Stalgast 620711 laser thermometer, Poland) before blood sampling. Around 10 mL of venous blood was placed in a test tube containing a clotting activator. The serum was separated from blood cells by centrifuging at 2000 rpm for 10 min. Selected biochemical parameters were determined in the separated serum. A 2 mL blood sample was placed in an EDTAK2 blood collection tube for the determination of hematological parameters. The acid-base balance was determined in a lithium heparin tube containing 2 mL of blood.

The concentrations of glucose, total cholesterol, high-density lipoprotein (HDL), and low-density lipoprotein (LDL), triacylglycerols, uric acid (UA) and lactic acid (LA) were determined in the Accent-200 biochemistry analyzer (*p*.Z. Cormay S.A., Poland) with the use of dedicated reagents. Colorimetric analyses were carried out at the wavelengths specific for each measured parameter.

Hematological parameters were determined in an ADVIA 2120i analyzer (Siemens, USA) which uses a combination of laser light scatter and cytochemical staining. Dedicated Siemens reagents were used in analyses of red blood cell (RBC) and platelet (PLT) counts, in colorimetric hemoglobin assay kits, and in PEROX and BASO assay kits for leukocyte differentiation.

Acid-alkaline balance values and oxygen consumption (in whole, venous and heparinized blood) were determined in the Rapid Lab 348 analyzer (Siemens, Oakville, ON, Canada) with ion selective electrodes. Siemens 6.8/7.3 buffer solutions were used in the reactions. In tests involving different analyzers, the coefficient of variation (CV) for the examined parameters was determined in the range of 0.03–2.42 in Rapidlab 348, 0.62–3.27 in Accent-200, and 1.7–12.9 in ADIVA 2120i.

### 2.4. Statistical Analysis

Basic descriptive statistics (mean, standard deviation–SD, and range of variation) and the normality of distribution (asymmetry coefficient, As) were calculated for each of the examined variables. All tested parameters had normal distribution, and the Student’s *t*-test for dependent samples was used to assess the significance of differences between the arithmetic means before and after sauna.

## 3. Results

### 3.1. Physiological Parameters during Sauna

The participants’ age, PA levels and physiological parameters during 66 min of sauna (66-ME) are presented in Table 1. The subjects’ average PA level was classified as moderate (1322 MET) based on IPAQ results. The mean values of HR_avg_ (102.5 bpm) were within the easy effort range, and the peak values of HR_peak_ (143.3 bpm) were within the very difficult range. Average energy expenditure was 486 kcal, and recovery time was estimated at 5 h. The estimated mean and peak values of VO_2avg_ and VO_2max_ (13.6 and 29.1 mL/min/kg, respectively), as well as EPOC_avg_ and EPOC_peak_ (9.8 and 20.4 mL/kg, respectively) were relatively low. The average respiratory rate (RR_avg_) was 18.5 breaths/min, and the peak respiratory rate (RR_peak_) was 29 breaths/min. Based on the adopted physiological criteria, the participants remained within the easy effort range for 2329.2 s, the moderate effort range for 1001.5 s, and the maximum effort range for only 35 s during the entire study (66-ME) (Table 1).

### 3.2. Physiological Parameters before and after Sauna

The changes in the values of selected physiological parameters (temperature, BP, and HR) before and after 72-ME are presented in Table 2. The mean BP values (systolic blood pressure–SBP, and diastolic blood pressure–DBP) decreased significantly (*p* < 0.001) from 134.6 to 124.9 mmHg (difference of 9.7 mmHg) and from 78.2 to 71.3 mmHg (difference of 6.93 mmHg), respectively, whereas HR values increased significantly (*p* < 0.01) from 74.6 to 82.0 bpm (difference of 7.4 bpm). Body mass decreased by 1.5 kg, and BMI decreased by 0.47 kg/m^2^, which can be attributed mainly to water loss caused by profuse sweating and increased lung ventilation (Table 2).

Blood morphological parameters before and after 72-ME are presented in Table 3. Significant quantitative and percentage changes were noted only in monocyte counts, which decreased from 0.47 (7.32%) to 0.42 (6.74%) after 72-ME.

Changes in body temperature, gasometric and biochemical parameters are presented in Table 4. A significant increase in partial pressure of oxygen (pO_2_; from 46.89 to 60.23; difference of 13.34; *p* = 0.030) and O2SAT values (from 73.2 to 82.1; difference of 8.9; *p* = 0.031) were noted during 72-ME. The following parameters decreased significantly: pH (from 7.41 to 7.31; difference of 0.04; *p* = 0.001), actual bicarbonate (aHCO_3_; from 28.69 to 26.09 mmol/L; difference of 2.51; *p* = 0.005), standard bicarbonate -(sHCO_3_; from 26.65 to 24.38 mmol/L; difference of 2.51; *p* = 0.003), BE extracellular fluid (BE (ecf); from 4.59 to 1.48 mEq/L; difference of 3.11; *p* = 0.022), BE blood test (BE (B); from 2.79 to 0.57 mEq/L; difference of 2.22; *p* = 0.005) and carbon dioxide concentration (ctCO_2_; from 29.99 to 27.41 mmol/L; difference of 2.58; *p* = 0.005).

A significant (*p* < 0.001) decrease in blood glucose (from 4.71 to 4.15 mmol/L; difference of 0.56; *p* = 0.001) and LA levels (from 1.69 to 1.50 mmol/L; difference of 0.19; *p* = 0.036) were also noted during 72-ME. In turn, a significant increase was observed in total cholesterol (from 185.17 to 198.27; difference of 13.1; *p* = 0.008), HDL (from 58.03 to 65.37; difference of 7.34; *p* = 0.006) and LDL cholesterol concentrations (from 100.23 to 110.43; difference of 10.2; *p* = 0.007). Triacylglycerol (TG) concentrations decreased by 11.57 mg/dL, but the observed changes were not significant.

## 4. Discussion

This study was undertaken to examine the effects of four 12-min sauna sessions separated by 6-min breaks and ending with cold water immersion for 1 min, on the physiological parameters of young and physically active men who were regular sauna users. Based on a wide range of the examined physiological and biochemical parameters, an attempt was made to determine whether strong thermal stress induces changes in homeostasis in the studied men.

The majority of research studies on sauna use indicate that regular sauna use can be regarded as a form of passive heat therapy and a biological regeneration method that delivers health benefits [12,13,37,38]. Emerging evidence suggests that sauna bathing exerts a particularly positive influence on the cardiovascular system [2,39,40,41]. Cardiometabolic health outcomes have been linked with the sauna’s beneficial impact on circulatory and cardiovascular function. However, the specific cardiovascular adaptations resulting from long-term sauna bathing remain elusive [42]. Heart rate is one of the key physiological parameters that change significantly under the influence of thermal stress in a dry sauna. The HR increases significantly even during short stays in the sauna that do not exceed 20 min [21], and it is highly elevated when sauna bathing lasts 45 min and longer. These correlations were investigated by Pilch et al. [12] in a study of ten healthy men aged 25–28 years who attended three 15-min sessions in a dry sauna and a steam sauna, with 5-min breaks in between. During the breaks, the participants cooled off in a cold shower and then rested in a sitting position. Their HR values increased from 66.6 to 126.0 bpm in the dry sauna, and from 66.2 to 138.2 bpm in the steam sauna. Dry sauna bathing induced a greater decrease in body mass than the steam sauna (−0.72 vs. −0.36 kg, respectively). However, a higher increase in HR values was reported during steam sauna (38.8% and 21.2%, respectively). The HR increased by 59.4 bpm during dry sauna bathing and by 72 bpm during steam sauna sessions (*p* < 0.01). In the present study, significant changes in HR values were also observed (from 74.6 bpm before sauna to 82.0 bpm after sauna; *p* < 0.01), but these differences were not clinically significant and were substantially lower than the values reported by Pilch et al. [12]. These discrepancies could be attributed to differences in the research methodology, namely the high effectiveness of the cooling phase after sauna, because HR was measured immediately after the cooling phase. The above assumption was validated by considerable differences in the values of HR_avg_ measured before the end of the fourth sauna session (66-ME–102.5 bpm; Table 1) and after the fourth sauna session (72-ME–74.6 bpm; Table 2). The effect of cold water immersion on reducing cardiometabolic markers has also been demonstrated by other authors [43]. The relatively small increase in HR values can also be attributed to the fact that the participants were physically active individuals and had attended 10 sauna sessions (twice a week) before the study. These outcomes point to high exercise capacity and good adaptation to thermal stress in the sauna. A recent study by Kunutsor et al. [42] confirmed the combined effects of cardiorespiratory fitness and frequency of sauna bathing on cardiovascular functions. Other researchers also demonstrated that the cardiac load during sauna corresponds to a moderate physical load of 60–100 W [44].

In our study, only minor changes in venous blood gasometric and biochemical parameters were observed, which, combined with the clinically nonsignificant differences in HR and BP values, points to a relatively low physiological load during sauna and the participants’ high exercise capacity. An increase in pO_2_ and O_2_ SAT values after sauna is logical because thermal stress increases lung ventilation, lung perfusion and oxygen uptake. Blood glucose decreased due to a higher metabolic demand for glucose during the sauna. Lactic acid levels were within the norm both before and after sauna. The decrease in LA levels after sauna probably resulted from a higher metabolic rate, oxygen uptake and higher LA utilization in bodily tissues.

The acute effects of passive heat therapies on circulating blood-based markers of cardiovascular function have been rarely investigated, but a considerable decrease in circulating arterial endothelial and platelet-derived microparticles was reported in individuals exposed to passive heat stress for around 56 min [45]. In the present study, significant sauna-induced effects were not observed in the analysis of hematological variables, excluding monocyte counts. A minor (at the level of the second decimal place) but significant decrease was noted in monocyte counts, but this observation was not diagnostically relevant because the analyzed parameter was within the norm. In contrast, Laukkanen et al. [40] reported an increase in leukocyte counts (monocytes are the largest leukocytes) from 6.2 (1.6) to 6.8 (1.6) × 109/L, and this parameter peaked immediately after sauna (duration: 30 min; temperature: 73 °C; humidity: 10–20%). Hemoglobin and thrombocyte levels were also significantly elevated immediately after sauna relative to pre-sauna values.

The effects of repeated sauna use on the lipid profile of young men have been rarely explored in the literature. Researchers reported a small and nonsignificant increase in HDL cholesterol levels and a transient decline in triacylglycerol concentrations in 16 male subjects (20–23 years) after 10 sauna sessions (each session consisted of three 15-min stays in the sauna, with 2-min cool-down breaks in between; temperature: 90 ± 2 °C, humidity: 5–16%) [46]. The mean levels of total cholesterol, LDL and HDL cholesterol, did not differ from the reference values in the tested sample. On the contrary, significant changes in lipid profile variables (total cholesterol, HDL, and LDL cholesterol) were reported before and after sauna bathing (all *p* < 0.01). From a physiological perspective, the sauna-induced increase in total cholesterol, HDL and LDL cholesterol levels results from an increase in blood density caused by sweating and loss of body water. Sweating led to a significant loss of body mass (1.5 kg), and similar observations were made in our previous study [47]. The metabolic rate increased significantly during sauna, but significant changes in body temperature (measured on the forehead) were not noted. The above could be attributed to the high effectiveness of cold water immersion during breaks [37], as well as highly effective thermoregulatory mechanisms and effective adaptation to thermal stress in a dry sauna with low relative humidity (14–16%). This assumption was validated by the relatively high body mass loss resulting mainly from sweating-induced dehydration, which is the main mechanism of heat loss from the body (evaporation of secreted sweat).

A significant decrease was also observed in SBP and DBP values. Blood pressure decreases in response to vasodilation to increase the rate at which heat energy is supplied to the skin and evaporated. Similar observations were made by other authors, including Laukkanen et al. [40] who reported a significant decrease in blood pressure after sauna. The mean SBP decreased from 137 mmHg before sauna to 130 mmHg immediately after sauna and remained unchanged at 130 mmHg after a 30-min recovery period (*p* < 0.001). The corresponding values of DBP were 82, 75 and 81 mmHg (*p* < 0.001). In contrast, Ketelhut and Ketelhut [34] reported a continuous increase in the SBP and DBP values of 19 healthy adult volunteers during a 25-min sauna session (temperature: 93 °C, humidity: 13%), which suggests that different responses to thermal stress in a sauna are possible. The results of this study may provide additional insights for explaining the emerging associations between regular sauna bathing, adaptation to heat stress and reduced risk of cardiovascular and metabolic diseases. The present findings have numerous practical implications, in particular for persons who participate in sauna sessions lasting more than one hour. The study demonstrated that extreme thermal stress is relatively safe and does not disrupt homeostasis in physically active men who are regular sauna users (several times a week).

## 5. Strength and Limitations

A homogenous and carefully selected group of physically active men who regularly used the sauna was a strength of the study. The results obtained in this representative group of subjects can be compared with other studies investigating persons who participated in prolonged sauna sessions (longer than 60 min), but were characterized by a different lifestyle (e.g., different physical activity levels) and different physiological profiles (e.g., different concentrations of cardiometabolic markers). A wide range of physiological parameters were analyzed with the use of standardized high-quality equipment; therefore, the present study meets strict research criteria (validity, reliability and objectivity) and supports the formulation of conclusions. The study focused on the physiological responses of men, and further research is needed to examine the responses of women and other representative groups.

## 6. Conclusions

High thermal stress in the sauna did not induce significant changes in the physiological parameters of young men, excluding body mass which decreased significantly due to profuse sweating. These observations indicate that the studied subjects were characterized by effective thermoregulatory mechanisms that maintain homeostasis, high exercise capacity and good adaptation to heat stress in the sauna. No significant changes were noted in the participants’ body temperature or other physiological parameters, which suggests that the described research protocol (passive heat stress combined with cold water immersion) is safe and can be recommended to young and physically active men who use a dry sauna on a regular basis.

## Figures and Tables

**Table 1 ijerph-18-11503-t001:** Physiological parameters during sauna (66-ME).

Parameter	Mean	SD	Min–Max	As
**Age [years]**	22.67	2.02	19–26	0.12
**PA level [MET]**	1322.2	407.1	870.0–2045.0	0.05
**HR_min_ [bpm]**	67.73	11.62	43.0–87.0	−0.31
**HR_avg_ [bpm]**	102.50	13.54	78.0–131.0	0.05
**HR_peak_ [bpm]**	143.33	18.31	102.0–176.0	−0.27
**Recovery time [h]**	4.94	4.30	0.0–17.0	1.20
**Energy expenditure [kcal]**	485.90	159.41	233.0–799.0	0.20
**VO_2avg_ [mL/kg/min]**	13.63	4.03	7.0–23.0	0.26
**VO_2max_ [mL/kg/min]**	29.07	7.14	15.0–45.0	0.42
**EPOC_avg_ [mL/kg]**	9.83	8.95	1.0–38.0	1.86
**EPOC_peak_ [mL/kg]**	20.40	20.25	2.0–83.0	1.91
**RR_avg_ [breaths/min]**	18.47	2.57	15.0–25.0	0.51
**RR_peak_ [breaths/min]**	28.97	6.60	22.0–46.0	1.19
**Physical effort range**				
**Easy < 107 [bpm]**	2329.23	1036.51	260.0–3960.0	−0.10
**Moderate 107–124 [bpm]**	1001.47	493.08	0.0–1661.0	−0.65
**Difficult 125–141 [bpm]**	460.60	471.83	0.0–1464.0	0.98
**Very difficult 142–159 [bpm]**	133.67	219.28	0.0–804.0	1.82
**Maximal ≥ 160 [bpm]**	35.03	107.95	0.0–537.0	3.93
**Total [bpm]**	3960.00	0.00	3960.0–3960.0	0.58

Notes: PA–physical activity; MET–metabolic equivalent of task; HR_min_–minimal heart rate; HR_avg_–average heart rate; HR_peak_–peak heart rate; VO_2avg_–average oxygen uptake; VO_2max_–maximal oxygen uptake; EPOC_avg_–average excess post exercise oxygen consumption; EPOC_peak_–peak excess post exercise oxygen consumption; RR–respiratory rate; RR_peak_–peak respiratory rate, As–asymmetry coefficient.

**Table 2 ijerph-18-11503-t002:** Physiological parameters before and after sauna (72-ME).

Parameter	Before Sauna				After Sauna				Difference (A–B)
	Mean	SD	Min–Max	As	Mean	SD	Min–Max	As	*t* (*p*)
**Temperature**	36.53	0.19	36.0–36.8	−0.94	36.62	0.56	34.7–37.4	−2.59	*ns*
**SBP [mmHg]**	134.63	12.95	107.0–155.0	−0.13	124.93	12.04	105.0–156.0	0.48	−6.61 (<0.001)
**DBP [mmHg]**	78.20	12.05	53.0–113.0	0.43	71.27	9.76	50.0–91.0	−0.34	−4.85 (<0.001)
**HR [bpm]**	74.57	12.99	50.0–111.0	0.51	82.00	13.59	61.0–119.0	0.59	3.26 (0.003)
**Body mass [kg]**	86.62	13.02	58.7110.7	−0.29	85.12	12.92	57.5–109.4	−0.27	−20.04 (<0.001)
**BMI [kg/m^2^]**	26.66	3.76	17.4–34.1	−0.50	26.19	3.73	17.0–33.7	−0.51	−18.24 (<0.001)

Notes: SBP–systolic blood pressure; DBP–diastolic blood pressure; HR–heart rate; BMI–body mass index; ns–not significant.

**Table 3 ijerph-18-11503-t003:** Descriptive statistics of blood parameters before and after sauna (72-ME).

Variables	Before Sauna				After Sauna				Difference (A–B)
	Mean	SD	Min–Max	As	Mean	SD	Min–Max	As	*t* (*p*)
**WBC (10^3^/µL)**	6.48	1.24	4.45–9.96	0.91	6.29	1.18	4.42–8.82	0.24	*ns*
**Neutro (10^3^/µL)**	3.68	0.87	2.36–6.63	0.71	3.65	0.79	1.87–5.46	0.35	*ns*
**Neutro %**	56.87	7.80	43.3–73.0	0.30	58.32	7.39	41.6–76.4	0.05	*ns*
**Lymph (10^3^/µL)**	2.02	0.61	1.01–3.47	0.26	1.89	0.57	0.96–3.48	0.49	*ns*
**Lymph %**	30.95	6.65	18.1–43.1	−0.11	30.09	6.54	14.7–46.4	0.32	*ns*
**Mono (10^3^/µL)**	0.47	0.12	0.23–0.76	0.48	0.42	0.11	0.19–0.75	0.74	−2.84 (0.008)
**Mono %**	7.32	1.72	3.9–11.2	−0.06	6.74	1.52	2.6–10.1	−0.35	−2.74 (0.010)
**Eos (10^3^/µL)**	0.15	0.10	0.03–0.42	1.01	0.15	0.11	0.02–0.52	1.26	*ns*
**Eos %**	2.32	1.30	0.5–5.2	0.54	2.33	1.44	0.4–6.5	1.17	*ns*
**Baso (10^3^/µL)**	0.03	0.01	0.01–0.06	0.17	0.04	0.02	0.01–0.07	0.52	*ns*
**Baso %**	0.53	0.17	0.2–0.9	−0.01	0.57	0.26	0.2–1.4	1.26	*ns*
**Leu (10^3^/µL)**	0.14	0.05	0.06–0.25	0.46	0.13	0.04	0.05–0.21	0.43	*ns*
**Leu %**	2.06	0.60	0.6–3.3	−0.27	1.95	0.55	0.6–2.8	−0.35	*ns*
**RBC (10^6^/µL)**	5.20	0.32	4.58–6.08	0.75	5.27	0.44	4.59–6.71	1.21	*ns*
**HGB (g/dL)**	15.53	0.91	13.9–17.5	0.23	15.72	1.23	13.9–20.4	1.26	*ns*
**HCT %**	44.09	2.45	39.0–48.8	−0.01	44.59	3.15	40.1–55.8	1.24	*ns*
**MCV (fl)**	85.0	4.60	70.0–92.6	−1.12	84.77	4.66	69.6–91.5	−1.27	*ns*
**MCH (pg)**	29.97	1.76	23.2–32.0	−1.81	29.92	1.85	23.4–32.1	−1.77	*ns*
**MCHC (g/dL)**	35.26	0.80	33.2–36.5	−0.54	35.28	0.77	33.6–36.8	−0.05	*ns*
**RDW %**	13.68	0.75	12.8–17.1	3.37	13.70	0.74	12.8–17.1	3.43	*ns*
**HDW %**	2.81	0.24	2.30–3.47	0.68	2.83	0.24	2.28–3.54	0.79	*ns*
**PLT (10^3^/µL)**	257.73	37.42	191–337	0.38	260.33	53.99	158–397	0.50	*ns*
**MPV (fl)**	7.83	0.66	7.0–10.1	1.36	7.81	0.60	7.0–9.5	1.01	*ns*

Notes: *ns*–not significant.

**Table 4 ijerph-18-11503-t004:** Descriptive statistics of venous blood gasometric and biochemical parameters before and after sauna (72-ME; N = 30).

Variable	Before Sauna				After Sauna				Difference (A–B)
	Mean	SD	Min–Max	As	Mean	SD	Min–Max	As	*t* (*p*)
**pH**	7.41	0.05	7.33–7.52	0.51	7.37	0.04	7.31–7.44	−0.24	−3.68 (<0.001)
**pCO_2_ [mm Hg]**	46.22	5.41	36.3–57.3	0.12	46.02	6.27	35.7–58.1	−0.06	*ns*
**pO_2_ [mm Hg]**	34.70	5.58	20.3–45.9	−0.85	41.83	5.28	30.4–57.2	0.35	7.13 (<0.001)
**O_2_ SAT [%]**	73.20	17.78	30.9–98.4	−0.38	82.10	15.71	30.9–98.6	−1.41	2.27 (0.031)
**aHCO_3_^−^ [mmol/L]**	28.60	4.47	23.1–39.6	1.13	26.09	2.30	22.9–31.3	0.34	−3.06 (0.005)
**sHCO_3_^−^ [mmol/L]**	26.65	3.88	21.7–35.7	1.21	24.39	1.25	21.7–27.5	0.83	−3.27 (0.003)
**BE (ecf) [mEq/L]**	4.59	6.36	−2.7–24.0	1.33	1.48	3.06	−2.0–12.1	1.98	−2.42 (0.022)
**BE (B) [mEq/L]**	2.79	4.30	−3.0–13.1	1.07	0.57	1.59	−1.8–4.0	0.60	−3.03 (0.005)
**ctCO_2_ [mmol/L]**	29.99	4.58	24.2–41.4	1.13	27.41	2.56	24.0–33.1	0.34	−3.02 (0.005)
**Na+ [mmol/L]**	142.50	3.19	137–150	0.53	142.93	3.79	135–151	−0.10	*ns*
**K+ [mmol/L]**	4.28	0.35	3.54–4.99	0.34	4.22	0.38	3.36–4.95	−0.57	*ns*
**Cl^−^ [mmol/L]**	105.78	2.08	102.1–110.0	0.18	106.20	1.95	102.1–110.1	0.53	*ns*
**Biochemical parameters**									
**Glucose [mmol/L]**	4.71	0.93	3.44–6.98	0.79	4.15	0.60	3.04–5.56	0.11	−4.42 <0.001)
**Total cholesterol [mg/dL]**	185.17	35.93	112–262	0.13	198.27	37.48	113–276	−0.05	2.87 (0.008)
**HDL [mg/dL]**	58.03	14.82	39–92	1.08	65.37	19.80	40–112	1.13	2.92 (0.006)
**LDL [mg/dL]**	100.23	29.51	35–175	0.72	110.43	29.22	45–182	0.33	2.89 (0.007)
**TG [mg/dL]**	123.97	75.94	56–341	1.59	112.4	73.24	33–357	1.73	*ns*
**Uric acid [µmol/L]**	335.47	61.50	222–497	0.73	340.67	67.08	227–521	0.87	*ns*
**Lactic acid (mmol/L)]**	1.69	0.37	1.09–2.65	0.58	1.50	0.28	1.02–2.26	0.41	−2.19 (0.036)

## Data Availability

The Excel data used to support the findings of this study are restricted by the Ethics Committee of the University of Warmia and Mazury in Olsztyn (UWM), Poland in order to protect participants’ privacy. Data are available from Robert Podstawski, E-mail: podstawskirobert@gmail.com for researchers who meet the criteria for access to confidential data.

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
