# Peer review of "The Effects of Repeated Thermal Stress on the Physiological Parameters of Young Physically Active Men Who Regularly Use the Sauna: A Multifactorial Assessment"

_ijerph, 2021, doi:10.3390/ijerph182111503_

Round 1

Reviewer 1 Report

The study presented is descriptive and includes numerous physiological measurements before and after the sauna in young physically active men. The study has numerous control variables at the methodological level.

It is a single study that shows how the sauna produces changes in certain physiological variables.

After careful review I find major weaknesses that make it unacceptable for publication in its present form. Major changes are required.

The main weakness of the study is the theoretical underpinning of the study to justify the objectives and hypotheses. In my view, the introduction is very generalistic and does not justify the enormous number of variables measured. The study should focus more on the specific variables, otherwise it remains a descriptive study. Furthermore, the usefulness of this objective should be clear and developed (what the article contributes on a practical level). All this also refers to the discussion, which is very focused on the significant results, but not on their meaning and usefulness.

On the other hand, it includes in the title that the participants were physically active young people, but then does not compare it with other types of populations. If this is an important aspect for the results it should be highlighted and argumented in the introduction and discussion.

Another flaw of the study, is the number of participants that it is very small, although being a repeated measures study reduces this limitation. However, it should appear in the limitations section of the study.

There is no limitations section in the discussion. There should be a critical review of the study by the authors.

One of the more important weaknesses of the study is the number of measures used. There is no justification for taking all available analyses and testing for differences between pre and post. You need to justify the use of one measure or another. Thus the number of measures included makes the introduction generalised and does not justify the measures used. In order to justify the objective of the study, there should be a literature review of the variables used, and in this study there is no such review; only general effects of the sauna appear. Therefore, the authors should specify the reason for each of the variables (or set of variables) in order to see the usefulness of the study. The same should be done in the discussion.

On the other hand, the inclusion of so many statistical contrasts detracts from the validity of the statistical significance, and should be adjusted for the contrasts.

And finally, throughout the manuscript I do not see the usefulness of the results. What is the benefit of knowing the changes produced after the sauna in terms of health, bearing in mind that it is a short-term stressor? How would these changes affect people's health in the long term? And specifically, what is the benefit of the subjects in this study being physically active?

In summary, the paper has some important weaknesses, but mainly it gives the impression that measures have been taken and tested for differences between them, without taking into account previous hypotheses and why these measures were taken. This is the biggest weakness of the article, making it descriptive and not focused on answering a research question. Therefore, I believe that the authors should make an effort to focus the study on specific variables based on the literature and their functionality on health, in the short and long term.

Author Response

Dear Reviewer,

Our response to Reviewers' comments is attached on file.

Best Regards

Reviewer 2 Report

The manuscript is not ready for publication. Throughout the text there are frequent errors that makes it unclear, and is disturbing to the reader. Many abbreviations are not explained. I will not mentions them separately. Some of them are well established, red blood cells (RBC) as example, but explanation of abbreviations has to be presented systematically. Must be checked across the complete text as well as the Tables.

The Introduction section is quite OK, but why do you contradict yourself by writing that studies have been performed in many countries, in many different age groups (lines 46-50), and then on line 81 stating that there is a "scaricity"?

The Research protocol is not informative enough. I do suggest a "time-line" as an illustration. The minute-descriptions are clear and fine - but what does ME stands for?

How was oxygen consumption measured?  A telemetry heart rate monitor is mentioned but that is not equal to VO2 or energy expenditure. This must absolutely mad more clear!

Is "pulsometer" an O2 saturation detection device? Which principle of detecting arterial O2 saturation (if so)? Brand, distributor etc is missing

Which cut offs were used for the different intensity levels? "Difficult" is not a correct description, I assume you mean "hard" or "high". Check common physical activity papers to learn how to use the English key terms. I do think you have them in reference 25 and 26!

On line 149 you start with blood samples. Continue with body temp, and then back to blood sampling again. Do organize your text in a more logical order.

What precision values do you have for your chemical analyses? Coefficient of variation at a specific concentration, for example?

There is no meaning in giving MET and kcal with decimal commas. That precision is not feasible to achieve.

Some text is repeating the content of the Tables, and a lot of different fonts and sizes are mixed in the text. All this disturbs the reading and gives the impression of an unfinished text, submitted without being checked by the authors.

Author Response

Dear Reviewer,

Our response to Reviewers' comments is attached on file.

Best regards,

Round 2

Reviewer 1 Report

Thank you for clarifying and explaining all my concerns. 

Author Response

Dear Reviewer,

Attached you will find second-round response.

Cincerely,

Reviewer 2 Report

Some improvements have been done but most of the comments from the reviewers have been rebutted.

Why not consider the suggestions more seriously rather than finding smart ways to contradict them?

I suggest you work them through again.

Author Response

Dear Reviewer,

Attached you will find our response to your comments.

Sincerely,
